# A Decade of Protecting Insect Biodiversity: The Impact of Multifunctional Margins in an Intensive Vegetable System

**DOI:** 10.3390/insects16020118

**Published:** 2025-01-24

**Authors:** Francisco Javier Peris-Felipo, Fernando Santa, Oscar Aguado-Martin, Ana Lia Gayan-Quijano, Rodrigo Aguado-Sanz, Luis Miranda-Barroso, Francisco Garcia-Verde

**Affiliations:** 1Syngenta Crop Protection, Rosentalstrasse 67, 4058 Basel, Switzerland; fernando.santa@syngenta.com (F.S.); ana_lia.gayan@syngenta.com (A.L.G.-Q.); 2Andrena Iniciativas y Estudios Medioambientales S.L., calle Gabilondo 16bis, 47007 Valladolid, Spain; oscaraguado@lepidopteros.com (O.A.-M.); rodrigoaguadosanz@gmail.com (R.A.-S.); 3Agricultura Sostenible Syngenta España, Calle de la Ribera del Loira, 8, 10, 28042 Madrid, Spain; luis.miranda@syngenta.com (L.M.-B.); francisco.garcia_verde@syngenta.com (F.G.-V.)

**Keywords:** biodiversity, vegetables, sustainability, habitat management, insect conservation, floral margins, population dynamics

## Abstract

The agricultural intensification over the last 80 years has led to the creation of large-scale crop fields and the loss of ecological elements, affecting natural communities. The implementation of measures such as floral margins allows for the creation of refuges for insect communities. The present long-term study (2013–2022) demonstrates that the sustained implementation of floral margins can effectively protect insect communities in intensive agricultural areas, highlighting their importance as a tool for fostering insect biodiversity.

## 1. Introduction

For decades, the intensification of agriculture has been imposing increasing pressure on biodiversity in agroecosystems. The range and abundance of thousands of plant and animal species have been in serious decline [1,2,3]. Consequently, ecosystems services have been decreasing over time [3,4,5], ultimately resulting in unsustainable agricultural production and even soil degradation [6,7,8,9].

Today, agriculture is practiced on roughly 50% of the usable land of planet Earth, making it one of humanity’s large impacts on the environment [10]. It is evident that effectively safeguarding our environment and preserving biodiversity is nearly impossible without incorporating agricultural landscapes into conservation efforts. Supporting and promoting biodiversity within agricultural landscapes is crucial both for the conservation of ecosystems and related services such as pollination, predation, or carbon sequestration. A well-established method proposed to protect biodiversity in agricultural systems is sowing crop field margins with wildflower mixes [11,12,13,14,15,16,17].

The use of margins, whether natural or implemented, based on seed mixtures of autochthonous species, appears to function as ecological corridors, linking isolated habitat patches and reducing landscape fragmentation [18,19]. Moreover, increasing the abundance of wildflowers, insects, and birds has been highlighted as an important way of promoting ecosystem services and supporting biodiversity conservation [20,21,22,23,24,25,26].

However, most studies on agroecosystems are based on short-term observations (1–3 years) [27,28,29] and are mainly focused on bees or bumblebees [18,30,31,32,33,34,35,36,37,38], with only a few considering other groups of insects such as beetles, butterflies, or hoverflies [28,35,39,40,41].

This work assumes that biodiversity can be assessed by measuring the abundance of insects and their presence in different environments in the long term, evaluating whether changes induced by floral margins are permanent over time. This leads to the testing of two hypotheses. First, there is a benefit of integrating floral margins to protect biodiversity. Second, the use of floral margins improves biodiversity over time. These hypotheses were tested on an intensive vegetable farm in Spain.

## 2. Materials and Methods

### 2.1. Areas of Study

The study was carried out on one highly productive Spanish vegetable farm located in Águilas (Murcia; 37°25′01.6″ N 1°36′13.7″ W) (Figure 1). The location area has a semi-arid Mediterranean climate [42] with hot summers (27.7 °C) and mild winters (13.7 °C) and with an average annual rainfall of 303 mm. Appendix A summarises the annual rainfall and temperature data recorded in the study area.

During the sampling period, the crops were rotated successively, leaving a fallow period between July and October. Crop rotation included: beet (Amaranthaceae; *Beta vulgaris* L.), celery (Apiacea; *Apium graveolens* L.), lettuce “romana” (Asteraceae; *Lactuca sativa* L. var. *longifolia*), lettuce “iceberg” (Asteraceae; *Lactuca sativa* L. var. *capitata*), lettuce “mini romana” (Asteraceae; *Lactuca sativa* L.), lettuce “baby gem” (Asteraceae; *Lactuca sativa* L.), onion (Amaryllidaceae; *Allium cepa* L.), and triticale (Poaceae; *Triticosecale* Wittm. ex A. Camus). All crops were planted in a design where the planting distance was 26 cm between the rows and 20 cm between the plants of a same row, except for onion, where the distance was 15 cm between rows and 15 cm between plants, and triticale, where the distance was 15 cm between rows and 3–6 cm between plants. The field size was 7.5 ha.

During the study, the growers stuck to their preferred agricultural practices, such as tillage, sowing, and fertilisation. Moreover, they continued with their same phytosanitary treatments as before, applying the appropriate products according to pest and disease thresholds. Any management measures were confined to the crop to avoid interference with the multifunctional margin.

### 2.2. Floral Margins and Plant Mixture Selection

The selection of plant species was based on several fundamental criteria such as the strict use of native species, ensuring a smooth climatic adaptation; not become a potential weed for the crop; featuring easy maintenance and capacity for self-sowing, as well as staggered flowering phenologies; and finally, being attractive for pollinators and natural enemies.

We established a floral margin using a herbaceous mixture consisting of *Borago officinalis* L. (7%) (borage; Fam. Boraginaceae), *Calendula officinalis* L. (17.5%) (pot marigold; Fam. Asteraceae), *Coriandrum sativum* L. (10%) (coriander; Fam. Apiaceae), *Diplotaxis catholica* (L.) DC. (5%) (wall-rocket; Fam. Brassicaceae), *Echium vulgare* L. (5%) (viper’s bugloss; Fam. Boraginaceae), *Lobularia maritima* (L.) Desv. (5%) (sweet alyssum; Fam. Brassicaceae), *Melilotus officinalis* (L.) Pall. (12.5%) (sweet yellow clover; Fam. Fabaceae), *Nigella damascena* (L.) (5%) (love-in-a-mist; Fam. Ranunculaceae), *Phlomis purpurea* L. (3%) (Jerusalem sage; Fam. Lamiaceae), *Salvia verbenaca* L. (10%) (wild clary; Fam. Lamiaceae), *Silene vulgaris* (Moench) Garcke (10%) (bladder campion; Fam. Caryophyllaceae), and *Vicia sativa* L. (10%) (common vetch; Fam. Fabaceae). The floral field margin of 3 m width × 300 m length was sown next to the crop area at 3 m from the field to facilitate daily work in the crop. Sowing took place using a seed electric drill with air distribution (APV 100 pneumatic, APV Technische Produkte GmbH, Hötzelsdorf, Austria) after the soil had been prepared with a flail mower. The seeds were covered using a rake. The seed sowing rate applied was 15 kg/ha. The field margin was mowed in autumn and then left to regrow the following season. During the first two years, supplementary seeds were added annually in March and April to ensure a consistent plant emergence by following this planting schedule; irrigation or watering was not necessary as the rainfall provided favourable growth conditions, as these species are adapted to the climatic conditions of the area.

### 2.3. Experimental Design and Sampling

To investigate the dynamics of effects of floral margin on insect biodiversity, the experiment was conducted over a period of 10 years (2013–2022). Insect abundance was assessed visually (flower observation) and by using sweeping nets (observed and captured specimens were merged to perform the corresponding analyses). The samplings were carried out one day per month between March and July by moving in a zigzag along 4 fixed transects of 50 × 2 m during 15 min per line and 4 times per day to avoid the light and temperature gradient and obtain a more representative sample. Replication became unfeasible due to the impossibility of locating uniform fields as constant crop rotations were dictated by market demands.

The collected specimens were introduced in a bottle with small amount of cyanide to keep them intact and to avoid discoloration. All specimens were identified to the genus or species level using appropriate entomological literature (see [43,44,45,46,47,48,49,50,51,52,53,54,55,56,57]). Specimens are deposited in the entomological collection of the National Museum of Natural Sciences (Madrid, Spain; MNCN).

### 2.4. Statistical Data Analysis

We implemented an approach based on count data regression modelling to study the temporal dynamics of the number of species and number of insects as a measure of their diversity. To accomplish that, we initially perform an exploratory data analysis of the diversity indexes, Shannon’s H, species richness, and Pielou’s evenness, comparing them among *orders* and across *years* to study their behaviour and detect patterns. We secondly characterise the temporal autocorrelation structure of the number of species and insects by building the correlation matrix of the yearly numbers to identify temporal autoregressive effects. We then propose a generalised linear mixed model (GLMM) for count data to describe the number of species and the number of insects. Thus, the model assumes that the number of species or insects follows either a Poisson distribution when its conditional mean and conditional variance are equal or a negative binomial when its conditional variance is greater than its conditional mean (overdispersion). The model is specified for the number of species and for the number of insects in the Equations (1) and (2), respectively.(1)$$\mu_{i,t}=\exp\left(\alpha +\beta t+\gamma_{i}+\gamma_{i}t+\lambda y_{ik,t-1}\right)$$(2)$$\mu_{ij,t}=\exp\left(\alpha +\beta t+\gamma_{i}+\gamma_{i}t+\delta_{j}+\lambda y_{ik,t-1}\right)$$

In Equation (1),
$\mu_{i,t}$
represents the conditional mean of the number of species for the
ith Order for a specific year t. In Equation (2), $\mu_{ij,t}$ represents the conditional mean of the number of insects for the ith Order, the jth specie, for year t. To account for temporal variation of the counts, the log-linear predictor incorporates an overall linear trend βt (fixed effect), a $\gamma_{i}$ term to represent the Order (fixed effect), a $\delta_{j}$ term for the Specie (random effect), and a interaction $\gamma_{i}t$ Order–year (fixed effect). On the other hand, the component λ captures the impact of the past of the process $y_{ik,t-1}$ in its future. The parameters of the models in Equations (1) and (2) are estimated via maximum likelihood and assuming two possible distributions for the response variable. The fitted models are compared to choose the best model to explain the variability of the counts by using likelihood measures and information criterions (AIC, BIC). All statistical data analyses are conducted in R statistical software by using the lm4 package (version 1.1-35.5).

The choice of generalised linear mixed models (GLMMs) for our analysis was driven by the complex nature of our ecological data and the specific research questions we aimed to address. GLMMs are particularly well suited for analysing count data in ecological studies as they can accommodate non-normal error distributions and account for both fixed and random effects. In our case, the use of GLMMs allowed us to model the discrete, non-negative nature of species and insect counts while accounting for the hierarchical structure of our data (species nested within orders).

The inclusion of temporal components and autoregressive terms in our models was crucial for capturing the dynamic nature of insect populations over time. Ecological systems often exhibit temporal dependencies, where the state of the system at one time point influences future states [58]. By incorporating a linear time trend (βt), we could model overall temporal changes in species and insect abundance. The interaction term between order and time $\left(\gamma_{i}t\right)$ allowed us to capture order-specific temporal trends, addressing potential differences in how various insect orders respond to environmental changes over time.

The autoregressive component $\lambda y_{ik,t-1}$ was included to account for temporal autocorrelation in our data. This term captures the influence of population sizes in the previous year on current year populations, a common phenomenon in population dynamics. While more complex time series models like ARIMA could have been considered, the relatively short duration of our study (10 years) limited their applicability. Our approach of incorporating autoregressive terms within the GLMM framework provides a robust alternative that accounts for temporal dependencies while allowing for the inclusion of other important predictors and random effects.

To address our research questions, we analysed the models described in Equations (1) and (2), with particular focus on the parameter β. The statistical significance and magnitude of β provide insights into the temporal effects on biodiversity. A statistically significant β indicates a meaningful change over time in either the number of species or the number of insects. Specifically, a positive β (β>0) suggests an improvement in biodiversity metrics over time in the presence of floral margins. However, to fully answer our research questions, we also considered the interaction between time and the presence of floral margins $\left(\gamma_{i}t\right)$. This approach allows us to assess both the overall temporal trends and the specific impact of floral margins on biodiversity enhancement over time. This modelling approach strikes a balance between model complexity and biological realism, allowing us to address our research questions while accounting for the inherent structure and temporal nature of our ecological data.

In order to complement the diversity analyses and inquire into community structure, log-series, log-normal, and broken-stick models were also applied [59]. The log-series model represents a community composed of a few abundant species and a high number of rare species. The broken-stick model refers to the maximum occupation of an environment with equitable sharing of resources between species. Finally, the log-normal model reflects an intermediate situation between the two [59]. Each of these models was applied to data obtained from the farm to calculate the expected number of species and the *log2* grouping of species according to abundance [59,60,61]. To test the significance of the model outputs, the expected species values were compared with those of the observed species through a chi-square analysis [62].

## 3. Results

### 3.1. Exploratory Data Analysis

Table 1 summarises the identified number of species and insects and their rate of change over the years. A total of 172 species belonging to seven orders, Coleoptera (24), Diptera (31), Hemiptera (3), Hymenoptera (71), Lepidoptera (41), Neuroptera (1), and Odonata (1), were captured during the ten-year research program. Appendix B compiles the list and abundance of each species captured during the study. Figure 2 shows the scatterplot of the counts of species and insects across the years and by orders. These results show that while the number of species increases during the first three years and then stabilises, the number of individuals approximately follows a linear growth trend over the analysed period. The population dynamics also exhibits that the most frequent order is *Hymenoptera*. The number of species and insects of this order doubles over almost the whole period in comparison with other orders.

Figure 3 presents the temporal evolution of the indices of species diversity, -diversity, Shannon’s Index *H*, species richness, and Pielou’s evenness *J*, by orders. The estimated values for the -diversity indexes show that *H*-index and richness exhibit the same behaviour: For all orders examined, the values approximately increase during the first three years and then stabilise. However, the increase seems to be higher for Hymenoptera and Lepidoptera than for Coleoptera and Diptera. On the other hand, Pielou’s evenness J shows differences between the orders. While in Hymenoptera and Coleoptera, the J-index follows the same pattern, with values increasing during the first three years and then stabilising; it shows a roughly linear decrease trend over the years for Diptera and almost constant values for Lepidoptera during the entire study period.

Figure 4 shows lower triangular matrices with correlation coefficients for the number of species and insects between years. Additionally, Figure 4 only lists statistically significant correlations at the significance level of 5%. Where no value is displayed, the associated correlation coefficient is not significant. The temporal pattern of the number of species is more stable. There is less dynamics across the years because the correlation coefficients between consecutive years are low and closer to zero. This may reflect the fact that the number of identified species is approximately constant over the study period. On the other hand, the temporal pattern of the number of insects is more dynamic across the years because the correlation coefficients between years are high and all are statistically significant. Moreover, in most of the cases, the correlation coefficients for the number of insects are higher between successive years than between non-consecutive years. Finally, the insect population dynamics exhibits some degree of autoregressive effects, i.e., the past counts of insects explain how future counts will look like.

### 3.2. Statistical Modelling

For each response variable, i.e., the number of species or the number of insects, four models were fitted based on the Equations (1) and (2), respectively. The latter means that for each response variable, the fitted models were: (a) a *full model* if the associated counts follow a *Poisson* distribution, (b) a *reduced model*, dropping the interaction *Order:time*, assuming that the associated counts follow a *Poisson* distribution, (c) a *full model*, assuming that the associated counts follow a *negative binomial* distribution, and, (d) a *reduced model*, dropping the interaction *Order:time*, assuming that the associated counts follow a negative binomial distribution.

Table 2 presents the analysis of deviance table and the statistics of goodness of fit of the fitted models. The results show that in the case of the number of species, the best model is the *reduced model*, assuming a *Poisson* distribution for the response variable. It means that the count of the number of species does not show evidence that their mean is different than their variance, i.e., the population dynamics of the number of species across the years is stable and is explained by the order, which is the unique statistically significant parameter. On the other hand, in the case of the number of insects, the best model is the *reduced model,* assuming a *negative binomial* distribution for the response variable. It means that the abundance of insects is a process with high variability as the best model assumes that the variance of the number of insects depends on its mean. Additionally, for this model, it was identified that the count of insects in the previous year, the linear temporal trend, and the order are statistically significant parameters. Thus, the number of insects increases linearly across the study period with different starting points for the observed orders and depends on the previous state of the population.

Table 3 summarises the exponentiated estimated coefficients, relative risk (RR), and their confidence intervals for the two selected *reduced models* with regards to the number of species (the selected model for the number of species only includes the *order* as an independent variable) and the number of insects. For both models, the reference category was the order Coleoptera. In the case of the model for the number of species, all parameters are statistically significant, and the estimated relative risks are higher than one. This means that it is 31%, 194%, and 80% more likely to identify a specimen of the orders Diptera, Hymenoptera, and Lepidoptera than one of the order Coleoptera. In the same way, with regards to the model applied to the number of insects, it was found that having fixed all other independent variables, an additional year is associated with 15% more identified insects. On the other hand, fixing all other independent variables, an increase of one identified insect in a specific year implies an increase of 2% more identified insects in the following year.

### 3.3. Community Structure Models

Figure 5 illustrates the evolution of species abundance classes over the 10-year study period. Using a log2-based grouping method, we established seven distinct abundance classes based on the number of individuals per species. The results reveal that Class 1, comprising species with fewer than 2.5 individuals, initially increases but shows a declining trend over time. In contrast, the remaining classes exhibit progressive growth as the abundance of individuals per species increases. This growth is particularly pronounced in classes representing more than nine individuals per species.

Moreover, the data analysis shows a notable variation in the rate of change across different abundance classes, as illustrated by the dashed arrows in the Figure 5. This variation demonstrates three clear patterns: rapid flux in low abundance classes (Classes 2–3), representing species with relatively low abundance, which exhibits a markedly high rate of change; moderate transitions in middle classes (Classes 4–6), indicating a level of stability for species with intermediate abundance levels; and stability in the high abundance class (Class 7), comprising the most abundant species, which shows a very slow rate of change, meaning a high degree of persistence and stability for dominant species within the ecosystem.

The analysis of community structure models (Table 4) reveals that the observed community patterns align significantly with both log-series and log-normal distributions (*p*-value > 0.05). However, the community structure deviates significantly from the broken-stick model (*p*-value < 0.05). This pattern is indicative of an unstable community characterised by a small number of abundant species coexisting with a large number of rare species. Interestingly, these results suggest that habitat factors were not the primary determinants of community structure. This conclusion is supported by the observation that the sampling area exhibited highly specific floral and faunal compositions.

## 4. Discussion

The selection of the seed mixture for implementing floral margins plays a decisive role in the successful attraction of insects and speed of biodiversity protection [4,28,38,39,40,41,63,64,65,66,67,68]. According to our first working hypothesis, this significant association between plants and insects can increase the number of species and individuals over time. Several studies have highlighted the influence of floral margins on insect abundance and their role as conservation practices, mainly associated with pollinators such as bees, butterflies, or beetles [32,63,65,66,67]. Most of the research carried out so far focuses on studies covering just one year (growing season) [41,64,65,66,67,68,69,70] or at most three [38,41]. In contrast, the present work covers a period of 10 consecutive years. Comparing our results with those of three-year studies, we found similar growth patterns triggered by floral margins, with rising numbers for both the number of insect species and individuals. Looking at the data that our research generated in the first three years (2013–2015), there is both an increase in the number of species (116.67%) and an increase in the number of individuals (78.75%).

According to our second hypothesis, the use of floral margins improves biodiversity over time. However, the dynamics of change in the number of species tends to stabilise after the third year (the increase of species is 0.63% between 2016 and 2022, with a total increase of 138.80% between 2013 and 2022), while the number of individuals shows a linear growth trend over the 10-year period (the increase of individuals is 173.92% bet-ween 2016 and 2022, with a total increase of 403.33% between 2013 and 2022). These results fit very well with the dynamics of populations: The number of species remains flat after three or four years because no new or modified management practices were implemented at the level of both crop and floral composition due to the fact that the planted crops (celery, lettuce, onion) have practically very similar agricultural management and, moreover, none of them are insect-dependent [71,72,73]. In contrast, the analysis of population dynamics reveals a self-induced process: insect abundance over successive years is related to insect species presence and numbers in the previous year, having a consistent increase in the abundance of certain species. This gradient in change across abundance provides valuable insights into community dynamics. It suggests that rare species are more susceptible to fluctuations, possibly due to environmental changes or competitive pressures; intermediate abundance species show a balance between stability and responsiveness to ecological factors, while the most abundant species demonstrate resilience, maintaining their dominance over time. The only articles we have found are based on studies conducted outside the agricultural ecosystem, where several authors have observed similar compliance with log-series and log-normal models in cerambycids (Coleoptera) [74,75] and braconids (Hymenoptera) [76,77,78,79]. Additionally, Lima et al. [80] observed similar trends in their population dynamics and demographics of the northern short-tailed shrew year after year. These observations have important implications for understanding ecosystem stability, succession processes, and potential responses to environmental changes or management interventions.

Moreover, our study is the first long-term work to assess the effects of floral margins on insect diversity. However, extrapolating the information from Noordijk et al. [81], a study of the impact of flower margins’ age on different groups of ground-dwelling species, we can observe that there is a correlation with species abundance growth over time. Recently, Claire et al. [29] analysed the effects of flower margins’ age on pollinator abundance in Hungary and found that abundance was higher when margins were younger and lower in older margins. However, a comparison of our research with the former is not practicable as in our study the margins were partially re-sown to maintain a good floral diversity.

## 5. Conclusions

Our findings provide compelling evidence that the establishment of floral margins in agricultural landscapes significantly enhances biodiversity over time. Field margins sown with diverse plant mixtures serve a crucial dual purpose: they not only contribute to biodiversity conservation but also boost the abundance of both species and individuals. This positive impact is observed in the short term and, importantly, persists over longer periods.

Moreover, our research underscores the critical role of flower margins as an essential and enduring strategy for biological conservation and ecosystem enhancement. This approach is particularly valuable in intensively farmed areas, where biodiversity is often under significant pressure. The implementation of flower margins offers a practical and effective method to counterbalance the ecological impacts of intensive agriculture.

## Figures and Tables

**Figure 1 insects-16-00118-f001:**
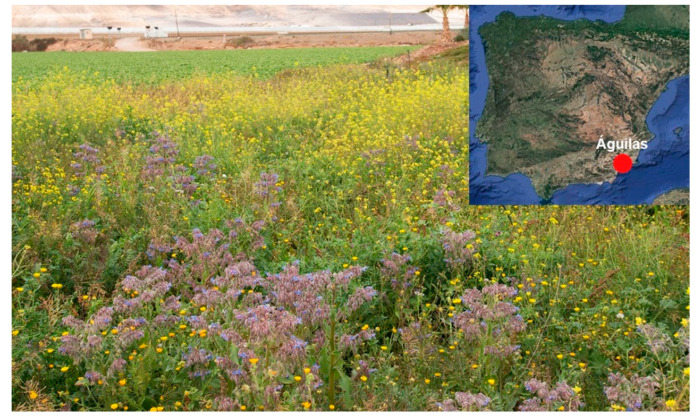
Floral margin on a farm in Águilas (Murcia) and its location in Spain.

**Figure 2 insects-16-00118-f002:**
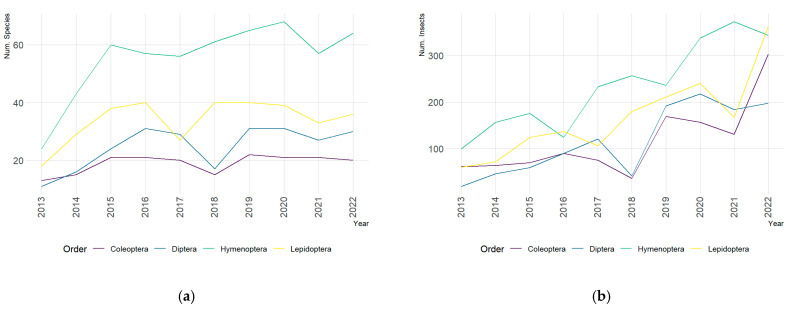
Scatterplot of the number of species and insects across the years. (**a**) Number of species. (**b**) Number of insects.

**Figure 3 insects-16-00118-f003:**
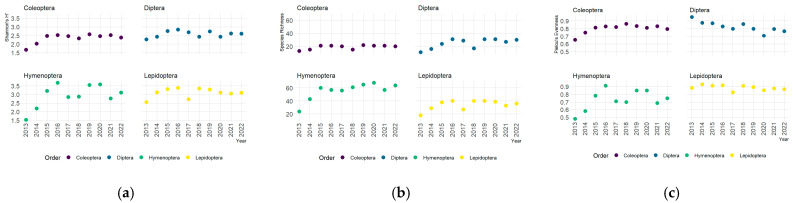
Scatterplot of the α-diversity indices: Shannon’s *H* (**a**), species richness (**b**), and Pielou’s evenness *J* across years (**c**).

**Figure 4 insects-16-00118-f004:**
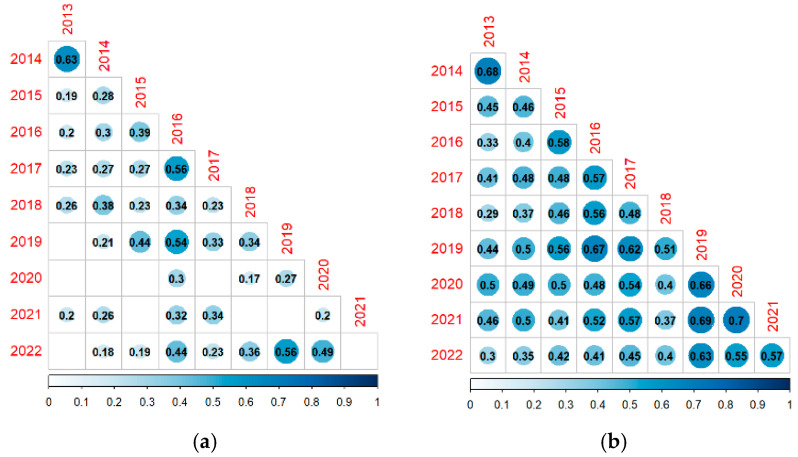
Correlation matrices of the number of species and insects between years. (**a**) Number of species. (**b**) Number of insects.

**Figure 5 insects-16-00118-f005:**
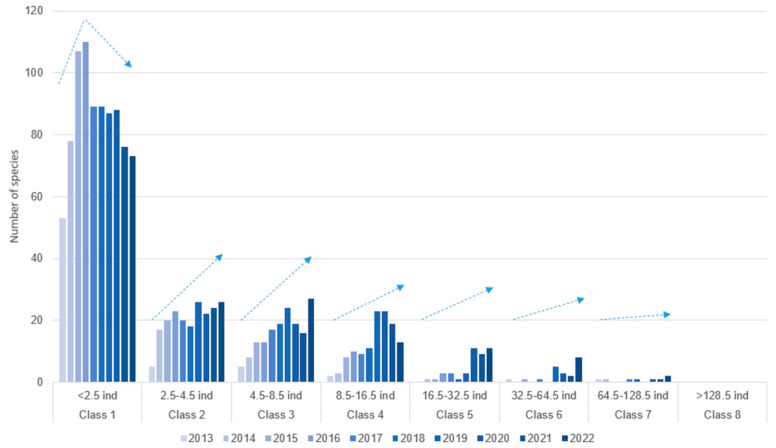
Changes of the species by abundance classes throughout the study (the arrows indicate the direction of change).

**Table 1 insects-16-00118-t001:** Number of species and insects and their rate of change across the years.

Year	Number of Species	Rate of Change (%)	Number of Individuals	Rate of Change (%)
2013	67		241	
2014	108	61.2	349	44.8
2015	150	38.8	443	26.9
2016	159	6.0	471	6.3
2017	140	−11.9	581	23.3
2018	139	−0.7	535	−7.9
2019	168	20.8	882	64.8
2020	167	−0.6	1038	17.7
2021	147	−11.9	907	−12.6
2022	160	8.8	1273	40.3

**Table 2 insects-16-00118-t002:** Analysis of deviance table and statistics of goodness of fit of the fitted generalised linear models and generalised linear mixed models for the number of species and insects, respectively (*** [0, 0.001]; ** [0.001, 0.01]; * [0.01, 0.05]).

*Number of Species*								
*Analysis of Deviance Table*								
Effect	Poisson		Poisson		Negative Binomial		Negative Binomial	
	Full		Reduced		Full		Reduced	
	LR Chisq		LR Chisq		LR Chisq		LR Chisq	
Lag num. Species	1.13		1.30		1.13		1.2976	
Order	30.76	***	30.76	***	30.76	***	30.76	***
Time	1.25		1.25		1.25		1.25	
Order:Time	0.85				0.85			
** *Statistics of Goodness of Fit* **								
AIC	229.3		224.1		231.3		226.1	
BIC	243.5		233.6		247.1		237.2	
Log.Lik.	−105.630		−106.057		−105.630		−106.057	
RMSE	4.22		4.3		4.22		4.3	
** *Number of Insects* **								
** *Analysis of Deviance Table* **								
**Effect**	**Poisson**		**Poisson**		**Negative Binomial**		**Negative Binomial**	
	**Full**		**Reduced**		**Full**		**Reduced**	
	**LR Chisq**		**LR Chisq**		**LR Chisq**		**LR Chisq**	
Lag num. Insects	0.41		0.13		22.53	***	23.30	***
Order	10.91	*	10.90	*	11.71	**	11.75	**
Time	804.31	***	812.50	***	238.09	***	237.30	***
Order:Time	31.56	***			3.06			
** *Statistics of Goodness of Fit* **								
AIC	6946.6		6971.9		6107.1		6104.1	
BIC	6999.5		7008.9		6165.2		6146.4	
ICC	0.7		0.7		0.5		0.5	
RMSE	4.85		4.95		11.48		12.58	

**Table 3 insects-16-00118-t003:** Exponentiated estimated regression coefficients and 95% confidence of the fitted generalised linear models and generalised linear mixed models for the number of species and insects, respectively (n.a.—not included because it was not statistically significant).

Parameter	Number of Species	Number of Insects
	RR (95% CI)	RR (95% CI)
(Intercept)	18.9 (16.33–21.72)	1.56 (1.09–2.24)
Order Diptera	1.31 (1.08–1.58)	0.67 (0.43–1.06)
Order Hymenoptera	2.94 (2.5–3.47)	0.56 (0.38–0.84)
Order Lepidoptera	1.8 (1.51–2.15)	0.88 (0.57–1.36)
Lag count	_	1.02 (1.01–1.02)
Time	_	1.15 (1.13–1.17)

**Table 4 insects-16-00118-t004:** Analysis of the community structure according to abundance models (log-normal, log-series, and broken-stick) for the insect community (*** [0, 0.001]; ** [0.001, 0.01]; * [0.01, 0.05]).

Community Structure Models											
Model	Year	2013	2014	2015	2016	2017	2018	2019	2020	2021	2022
Log-normal	Chisq	6.81	3.42	8.20	10.12 *	6.39	6.41	8.46	7.49	5.88	6.77
	*p*-value	0.234	0.635	0.145	0.034 *	0.380	0.260	0.132	0.277	0.436	0.342
Log-series	Chisq	9.12	4.260	3.238	3.208	4.013	3.420	4.288	3.263	2.565	2.611
	*p*-value	0.166	0.512	0.663	0.523	0.674	0.6355	0.508	0.775	0.861	0.855
Broken-stick	Chisq	21.212	18.464	17.809	17.832	20.313	15.750	20.595	27.982	23.284	32.623
	*p*-value	0.0007 ***	0.002 **	0.003 **	0.001 **	0.002 **	0.007 **	0.0009 ***	0.0001 ***	0.0007 ***	0.0001 ***

## Data Availability

The data presented in this study are available in Appendix B.

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
