# Peer review of "A Decade of Protecting Insect Biodiversity: The Impact of Multifunctional Margins in an Intensive Vegetable System"

_insects, 2025, doi:10.3390/insects16020118_

Round 1
Reviewer 1 Report
Comments and Suggestions for Authors
The authors conducted a 10-year study on insect fauna following the sowing of a floral margin in the begin of the experiment. This long-term monitoring of insect populations in response to habitat management is relevant, as few studies provide such extensive datasets. For this reason, I believe the study could be considered for publication. However, significant improvements are required before acceptance.
Overall, I have many concerns with data acquisition and data analysis
The authors express their main research questions at the end of the Introduction “First, there is a benefit of integrating floral margins to enhance biodiversity. Second, the use of floral margins improves biodiversity over time” However, the current statistical approach is inadequate to answer these questions. For instance, I do not understand why authors are interested in correlation between years (Fig. 4). Moreover, if the authors would be interested in studying time series patterns, they should use autoregressive models such as ARIMA. Yet, these models require longer time series (50+ time units) to study periodic patterns.
To analyse temporal evolution effectively, the authors could use the variable "time" as an independent variable. Testing different models (e.g., linear, logarithmic, or other functions) to identify the best fit for observed patterns over time would provide more robust insights. This could be done for each insect order, allowing for comparisons of response patterns and coefficients across orders. Rather than emphasizing whether orders differ in abundance (which is expected), the focus should be on how each order responds differently to habitat management over time. For example, richness and abundance trends (e.g., Fig. 2a, 2b) could be analyzed with simple models, such as linear or logarithmic regressions, using "time" as the independent variable, for each order.
Diversity analyses should be simplified to include richness and diversity indices while avoiding overly complex or redundant methods.
Specific comments:
Introduction- This is too short and does not provide a global overview of current knowledge.
Material and Methods (Lines 99–102): The sampling methodology lacks crucial details. Specifically, it is unclear where the transects were conducted: i) Were they located in the crop field or on the floral margin? ii) If in the crop field, how distant were the transects from the floral margin? This distinction is critical to understanding the potential spillover effects of insects from the floral margin to adjacent areas. Ideally, the authors should have done transects at different distances from the floral margin,
Statistical Data Analysis (Lines 108–138): The modelling approach is poorly justified and overly complex, with an excessive emphasis on models without proper hypothesis testing (see comments above)
Results Section: The results require reanalysis and reorganization.
Lines 178–193 and Fig. 4: These sections are problematic. Correlations between years lack relevance and should be removed. Autocorrelation analyses are unnecessary here, given the dataset's constraints.
Lines 195–233: Statistical analyses should focus on time-dependent models for each order separately, using time as a continuous variable, as outlined above.
Figure and Table Recommendations: Fig. 3: Richness data is redundant with Fig. 2b and should be removed. Table 5 is unnecessary and could be summarized in the text (ideally all this information could be removed). Overall retain a set of figures and tables that focus on the main findings.
The Discussion section requires revision following the reanalysis of the data. Greater emphasis should be placed on comparing the results with findings from other relevant studies. Several sentences should be toned down for accuracy (e.g., lines 300–301). Additionally, some statements lack clear relevance to this study (e.g., lines 296–298) and should either be clarified or removed to maintain focus and coherence.
Author Response
Dear reviewer,
Thank you for providing such a comprehensive review of our submitted manuscript. We value the time and expertise each of the four reviewers has contributed. To address your questions, suggestions, and concerns efficiently, we have classified and consolidated all comments, focusing particularly on those regarding the generalized linear mixed models and regression models. This approach allows us to respond comprehensively while ensuring all reviewers' points are addressed in context.
Most of the suggested comments have been implemented in the document. We sincerely appreciate the reviewers' comments on our statistical approach. The methods we used were carefully selected to address our specific research questions and to handle the complexities of a 10-year longitudinal dataset in an ecological context.
Kind regards,
Authors

Reviewer 2 Report
Comments and Suggestions for Authors
The authors have done a good job to document long term effects of floral field margins on insect biodiversity changes. This study is particularly important in the vegetable ecosystem. However, the authors need to put in a little more effort to make it a more attractive and citable paper.
Comments
Title: You might want to revise the title: Use the singular form for the landscape, as it is only one. I would suggest making it more specific as you only did it to vegetable intensive system.
Abstract: Specific details are missing in the abstract. You need to add some specific results. Moreover, remove the numbering from 1-5. Why it has been added here? Always remember abstract is a short version of your paper. So, make it like that!
L20: Revise good emergency. You can write it as consistent floral emergence or floral consistency.
L20-22: Replace it with specific results and rewrite for more clarity. The wrong phrases have been used which are much irrelevant to the scientific writing. Try to be specific and clear.
L22-23: Again, you are using too much general results. Come up with something very specific.
L23-24: Same comment.
Introduction:
L31-40: Merge these two paragraphs into a single paragraph because you are trying to build a case of intensive use of land for agriculture and the biodiversity loss. The flow is missing and there is a lot of redundant content. Use word approximately instead of roughly.
L38-40: I cannot understand this sentence. What are you trying to convey? Revise for grammar and clarity issues.
L42: When you say ecosystem services then also mention other services like predation, pollination, carbon sequestration, etc. Revise this sentence.
L 43: Use strengthen or improve instead of bolster.
L46-48: Revise this sentence for more clarity and be very specific when you use ecological terms. Nature conservation vs. ecosystem services.
L51-53: Revise this sentence. This sentence does not match with the hypothesis.
L53-56: Revise the grammar and write a single hypothesis, as both are same.
General comment for introduction section: The introduction section lacks a flow and details. I would recommend enhancing this section for clarity, specifically the paragraph formation. Pay special attention to the last paragraph.
Materials and methods
L67-68: In addition to common names, add scientific names, authority and family name of each vegetable.
L69-70: Is this the planting distance for all vegetables? If yes, then mention it.
L71-73: Details are missing here. Fertilizer application? Pesticide treatments? Others?
L76-79: This sounds like you have started discussion in materials and methods. Be specific. These sentences are unnecessary.
L84-89: Common name, authority and families are missing.
L91: Write the manufacturer, make, company, country of flail mower in brackets.
L91: What is a drag?
L92: What is sowing dose? Is it the seed sowing rate?
L93-94: Give a range of months, when you added the seed mix.
L99: Use sweeping instead of assessment.
L102-103: It is unclear what you want to express here? No replication? How?
L104-105: Insects are never preserved in cyanide. You might have used a killing bottle with a small quantity of cyanide added.
L106-107: Use correct tense here.
L108-149: How was over-dispersion tested and how did you choose Poisson or negative binomial models? Did you check for multicollinearity among predictors?
Results
L176: Figure 3: Just write A, B, C. You have mentioned their description in the caption.
L193: Revise Figure 4 for the x and y axes information. Also, in the 4b, y2013, y2014 looks odd. Try to be consistent. Remove y’s.
Tables general comment: Revise to follow the MDPI Insects template.
Discussion:
Overall, the discussion needs revision. You have not discussed the main results you got. You have not discussed why hymenopterans are increasing over time? What could be the implications of this increase? How could it improve the services in the vegetable ecosystem? Also add the limitations of your study. You have missed a major group, i.e., spiders, which are abundant in most of the ecosystems, and this is a limitation of your work. You have not covered the arthropods other than insects. Also, you have disturbed the floral strip through mowing operations. Now, this is another part which you need to mention. Add the future directions. What could be the practical implications? You did a 10-year study so your discussion should be well designed and should have more content.
Moreover, you have not cited a single recent study. Add following studies published in 2024:
Antkowiak, M., Kowalska, J., & Trzciński, P. (2024). Flower Strips as an Ecological Tool to Strengthen the Environmental Balance of Fields: Case Study of a National Park Zone in Western Poland. Sustainability, 16(3), 1251.
Haider, S., Khan, F. Z. A., Gul, H. T., Ali, M., & Iqbal, S. (2024). Assessing the role of conservation strips in enhancing beneficial fauna in the wheat-cotton agricultural system in Punjab, Pakistan. Pak. J. Zool, 56(6), 1-9.
Hamon, L. E., Kilpatrick, L. D., & Billeisen, T. L. (2024). The Impact of Wildflower Habitat on Insect Functional Group Abundance in Turfgrass Systems. Insects, 15(7), 520.
Mockford, A., Urbaneja, A., Ashbrook, K., & Westbury, D. B. (2024). Wildflower strips enhance pest regulation services in citrus orchards. Agriculture, Ecosystems & Environment, 370, 109069.
L495-496: Add family between the order and the species. This will improve the information presented in Appendix 2.
Follow the MDPI Insects format for reference styling. Multiple mistakes found.
Author Response
Dear reviewer,
hank you for providing such a comprehensive review of our submitted manuscript. We value the time and expertise each of the four reviewers has contributed. To address your questions, suggestions, and concerns efficiently, we have classified and consolidated all comments, focusing particularly on those regarding the generalized linear mixed models and regression models. This approach allows us to respond comprehensively while ensuring all reviewers' points are addressed in context.
Most of the suggested comments have been implemented in the document.
Kind regards,
Javi
Reviewer 3 Report
Comments and Suggestions for Authors
I have reviewed the manuscript titled "A Decade of Enhancing Insect Biodiversity: The Impact of Multifunctional Margins in an Intensive Agricultural Landscapes".
Nowadays, the topic of agricultural intensification and its impact on natural ecosystems has garnered significant attention from researchers in various countries due to its long-term destructive effects, as well as the emergence of various pests that, in turn, increase the costs of production. The findings of this research are particularly important in terms of a professional survey on biodiversity management, focusing on the cultivation and preservation of flowering plant species in proximity (margins of) to agricultural fields.
A noteworthy aspect of this manuscript is the results are obtained from long-term sampling over a period of 10 years, which minimizes potential errors by encompassing seasonal and inherent patterns of the ecosystem, thereby providing interesting results.
This article is expected to be frequently cited in subsequent research after publication and serve as an applied protocol by agricultural extension and environmental conservation institutions. Therefore, I recommend to accept after a minor revision.
Given the large volume of information and the broad scope of the background, the authors have excessively condensed the article's content, which may lead to a lack of clarity in certain sections for readers. I request that the authors carefully review the comments noted on the original file and add further explanations where necessary for each topic, as well as cite relevant references.
Best Regards

Author Response

(The authors gave the same response as above.)

Reviewer 4 Report
Comments and Suggestions for Authors
This article presents an interesting compilation of data over a long period of time, which is its main value from the point of view of scientific dissemination. I believe that the work can be published in Insect if a number of changes and improvements are introduced.
Introduction
L33: cites 1-3 do not correspond to the sentence. Please, replace with more suitable ones.
Include, in this section, information on previous results of the floral margin used that support the subsequent sentence (lines 76-77, M&M) "Floral margins based on seed mixtures of autochthonous species and planted in combined strips are the fastest way to provide significant biodiversity benefits within farmed…”
L53-56: Actually, I think the first hypothesis has not been tested in this paper because there is no data comparing the results with plots without margins. There is only a comparison of the results of the same plot in successive years (hypothesis 2).
Materials and Methods
L61-62: “with hot summers (27.7°C) and , mild winters (13.7°C) and with an 61 average annual rainfall of 303 mm
L76-78: These are statements that are best placed in the introduction or discussion sections and supported with citations.
L 84: What is the origin of the seeds?
L90-94: I am not clear about the time of sowing. Was there an initial sowing in spring and annual re-sowing? What exactly does supplementary seeds were added... mean? Was there irrigation?
Section 2.3: this section needs improvement. How many lines were included in the transect? How many samples per year? one or several? In what periods were they made? How was the visual sampling, was focus on the flowers?
Section 2.4:
add the exchange rate formula (see table 1)
Add the formulas for the species diversity indices and justify why the three are presented or select the most appropriate one and remove the rest. Figures 3a, 3b and 3c are very similar
Results
Figures 2 and 3 need to improve their quality in the final version
L221: Remove superscript 1 in species
L257-263: explain further what you mean, especially in relation to environmental factors (in this section or in the discussion section)
Discussion
The discussion of the results begins by talking about the importance of the selection of the seed mixture for the floral margin. However, this thesis is not developed. The authors should explain their experience with the mixture used and its main advantages and disadvantages.
L280: I do not agree that two hypotheses have been tested in this work (see comment L53-56)
L285-286 “The number of species remains flat after a few years because no new or modified management practices were implemented at the level of both crop and floral composition”, but there was a constant change of crops if I have understood M&M's explanation correctly, which would imply different agricultural practices.
L289-294: include very general and obvious statements from the ecological point of view. It would be desirable to illustrate this discussion with the adaptation to the agroecosystem of the rarest and most abundant specific species.
L297 public health or plant health?
L296-299 Is the observed process so unique? Why?
Explain why Hymenoptera were more abundant in number of species.
Author Response

(The authors gave the same response as above.)

Round 2
Reviewer 2 Report
Comments and Suggestions for Authors
The authors have incorporated all the suggested changes.
Reviewer 4 Report
Comments and Suggestions for Authors
The paper has improved substantially in its second version.